# FRACTIONAL GRAPH CONVOLUTIONAL NETWORKS (FGCN) FOR SEMI-SUPERVISED LEARNING

## ABSTRACT

Due to high utility in many applications, from social networks to blockchain to power grids, deep learning on non-Euclidean objects such as graphs and manifolds continues to gain an ever increasing interest. Most currently available techniques are based on the idea of performing a convolution operation in the spectral domain with a suitably chosen nonlinear trainable filter and then approximating the filter with finite order polynomials. However, such polynomial approximation approaches tend to be both non-robust to changes in the graph structure and to capture primarily the global graph topology. In this paper we propose a new Fractional Generalized Graph Convolutional Networks (FGCN) method for semi-supervised learning, which casts the Lévy Flights into random walks on graphs and, as a result, allows to more accurately account for the intrinsic graph topology and to substantially improve classification performance, especially for heterogeneous graphs.

## 1 INTRODUCTION

Adaptation of deep learning (DL) to graphs and other non-Euclidean objects has recently witnessed an ever increasing interest, leading to the new subfield of *geometric deep learning*. In particular, geometric deep learning is an emerging direction in machine learning which aims at generalizing the concepts of deep learning for data in non-Euclidean spaces, e.g., graphs and manifolds, by bridging the gap between graph theory and deep neural networks (see discussion by Bronstein et al., 2017; Monti et al., 2017; Monti, 2019, and references therein).

Many such DL approaches for non-Euclidian objects are based on the idea of performing a convolution operation in the spectral domain with a suitably chosen nonlinear trainable filter. As a result, node features are mapped into some Euclidian space. Next, graph filters are approximated with various finite order polynomials, e.g., Chebyshev polynomials (i.e., the ChebNet model family of Defferrard et al., 2016; Kipf & Welling, 2017), Cayley transform (i.e., CayleyNet of Levie et al., 2018) or the generalization of polynomial filters in a form of Auto-Regressive Moving Average (ARMA) models (Bianchi et al., 2019). However, deep learning approaches based on approximation with finite order polynomials tend to be non-robust to even minor changes in the graph structure and to largely disregard the local graph topology that often plays the critical role for learning on heterogeneous graphs. In contrast, as noted by Bianchi et al. (2019), one of the primary benefits of the ARMA filters over polynomial ones is that ARMA filters are not computed in the Fourier space induced by a graph Laplacian, and as a result, ARMA filters are local in the node space and enable to more flexibly and accurately capture the underlying graph topology.

In this paper we further advance this localized approach to DL on graphs and propose a fractional generalized graph-based convolutional filter for semi-supervised learning, which casts the Lévy Flights into random walks on graphs. As a result, our new Fractional Generalized Graph Convolutional Networks (FGCN) method allows to more accurately account for the intrinsic local graph topology and to substantially improve classification performance, especially for heterogeneous graphs.

The key contributions of our paper can be summarized as follows:

- we propose a new fractional generalized graph-based convolutional filter for semi-supervised learning, which casts the Lévy Flights into random walks on graphs and, as a result, provides a more efficient exploration of the graph structure.

- we develop a new Fractional Generalized Graph Convolutional Networks (FGCN) method that substantially improves accuracy of node classification for graphs of smaller sizes and with a lower number of node features.

- the proposed architecture of FGCN uses two state-of-the-art operations – gated max-average pooling and residual block. The architecture has been shown to significantly improve the training convergence and model output stability.

## 2 RELATED WORK

Many earlier semi-supervised learning approaches on graphs, e.g., Gaussian mixture models, co-training, harmonic function, and label propagation, employed only the label information (i.e., labeled instances) to train models based on the smoothness assumption over the labels (Zhu & Goldberg, 2009) and largely disregarded the graph structure. To improve performance, several learning methods on graphs propose to incorporate intrinsic "graph-based" information by designing a classifying function via generalizing the normalized cut and adding smooth function with respect to the intrinsic structure (Zhou et al., 2004; Zhou & Burges, 2007). A recent approach of Avrachenkov et al. (2012) proposes a generalized optimization framework through considering the above two methods as particular cases. However, one of the major limitations to these graph-based semi-supervised learning methods is disregarding important information contained in graph edges.

To address this disadvantage, Defferrard et al. (2016) propose a formulation of convolutional neural networks (CNN) based on spectral graph theory – ChebNet, which employs approximation via finite order polynomials and, in particular, is based in the Chebyshev expansion for fast filtering instead of the expensive eigen-decomposition. Graph Convolutional Networks (GCN) of Kipf & Welling (2017) simplifies ChebNet and further addresses the gradient vanishing problem and reduces the number of optimization. Other related approaches to graph learning with deep neural networks include, for instance, mixture model networks (MoNet) (Monti et al., 2017), graph attention networks (GAT) (Veličković et al., 2017), graph convolutional recurrent networks (Seo et al., 2018), dual graph convolutional networks (Zhuang & Ma, 2018), FastGCN (Chen et al., 2018), and simplified version of GCN (Wu et al., 2019).

To extend the success of GCN on undirected graphs to directed graphs, MotifNet of Monti et al. (2018) replaces the normalized Laplacian with the *motif Laplacian* in a multivariate polynomial filter, where the motifs information can help capture the network structure. Finally, the most recent approach of Bianchi et al. (2019) provides more flexible responses than GCN by using parallel and periodic concatenations of the convolutional kernel via the ARMA filter. As a result, the ARMA approach which is applicable to both directed and undirected networks allows to more accurately incorporate the underlying local graph structure into the graph learning process.

## 3 METHODOLOGY

Given a graph structure $\mathcal{G} = \{\mathcal{V}, \mathcal{E}, W\}$, with $\mathcal{V}$ the set of nodes ($N = |\mathcal{V}|$) and $\mathcal{E} \subseteq \mathcal{V} \times \mathcal{V}$ the set of edges between any two nodes. Let square matrix $W$ represents the adjacency matrix of a graph whose entries $\{\omega_{ij}\}_{1 \leq i,j \leq N}$ are the edge weight function such that each edge $e_{ij} \in \mathcal{E}$ has a weight $\omega_{ij}$. We call $W$ the symmetric adjacency matrix ($\omega_{ij} \neq 0$ if there is an edge between vertex $v_i$ and vertex $v_j$) if we assume that graph $\mathcal{G}$ is undirected. In reality, however, undirected graph is a simplified representation of complex networks, we also consider directed graph (in which case $W^\top \neq W$) and we can easily transform the non-symmetric $W$ to the symmetric one through $W^{'} = (W^\top + W)/2$.

We take $N \times Q$ feature matrix $X$ as the input to a semi-supervised learning algorithm where $Q$ is the number of different node features. Suppose we would like to classify $N$ data points into $K$ classes (communities), we define $N \times K$ label matrix $Y$ as:

$$Y_{ik} = \begin{cases} 1, & \text{if vertex } i \text{ is labelled as a class } k, \\ 0, & \text{otherwise.} \end{cases} \tag{1}$$

Here we refer to each column $Y_{\cdot k}$ of matrix $Y$ as a *labeling function*. Also define an $N \times K$ matrix $F$ and call its columns $F_{\cdot k}$ *classification functions*.

## 3.1 GRAPH SIGNAL PROCESSING

In graph signal processing, a real-symmetric matrix has $N$ real eigenvalues and its $N$ real eigenvectors form an orthonormal basis. Given the symmetric adjacency matrix $W$ of a graph, let $D$ be the degree matrix where $d_{ii} = \sum_{j=1}^{N} w_{ij}$ and denote $L = U^{\top} \Lambda U$ the Standard Laplacian matrix, where $\Lambda = diag(\lambda_0, \dots, \lambda_{N-1})$ and $U = [u_0, \dots, u_{N-1}]$ is the matrix of eigenvectors.

In the following, we will revisit three popular semi-supervised learning methods - graph-based semi-supervised learning, fractional graph-based semi-supervised learning, and graph convolutional networks and gain new insights for improving their modeling capabilities.

**Graph-based semi-supervised learning.** Graph-based semi-supervised learning (G-SSL) has received much attention as an alternative approach to the population paradigm of supervised learning in recent years. G-SSL develops a generalized optimization framework, which has three particular cases (i) the Standard Laplacian (SL); (ii) Normalized Laplacian (NL); (iii) PageRank (PR). The optimization formulation with the following expression:

$$\min_{F} \left\{ 2F_{\cdot k}^{T} D^{\sigma-1} L D^{\sigma-1} F_{\cdot k} + \mu (F_{\cdot k} - Y_{\cdot k})^{T} D^{2\sigma-1} (F_{\cdot k} - Y_{\cdot k}) \right\}, \tag{2}$$

where $\mu$ is a regularization parameter. The minimization of the first term in Eq. 2 corresponds to the idea that if two nodes are close in graph with respect to some metric, they should belong to the same class; and by minimizing the second term we would like to bring the classification function $F_{\cdot k}$ as close as possible to the labelling function $Y_{\cdot k}$. The Eq. 2 allows us to obtain the Standard Laplacian based formulation ($\sigma = 1$), the Normalized Laplacian formulation ($\sigma = \frac{1}{2}$), and PageRank formulation ($\sigma = 0$). The objective of the generalized optimization framework for G-SSL is a convex function and the corresponding classification function:

$$F_{\cdot k} = (1 - \alpha) \left( I - \alpha D^{-\sigma} W D^{\sigma-1} \right)^{-1} Y_{\cdot k}, \tag{3}$$

where $\alpha = \frac{2}{2+\mu}$ and for $k = 1, \dots, K$.

Tuning the parameter $\sigma$ on the power of degree matrix $D$, we can obtain three mentioned above particular semi-supervised learning methods:

- SL method ($\sigma = 1$): $F_{\cdot k} = (1 - \alpha) \left( I - \alpha D^{-1} W \right)^{-1} Y_{\cdot k}$

- NL method ($\sigma = \frac{1}{2}$): $F_{\cdot k} = (1 - \alpha) \left( I - \alpha D^{\frac{-1}{2}} W D^{\frac{-1}{2}} \right)^{-1} Y_{\cdot k}$

- PR method ($\sigma = 0$): $F_{\cdot k} = (1 - \alpha) \left( I - \alpha W D^{-1} \right)^{-1} Y_{\cdot k}$

From above formulations, the classification function $F$, i.e., the result of random walk process which provides the connection to the probabilistic interpretation of G-SSL. The parameter $\alpha$ controls the strength of the ground truth label matrix $Y$ in the generalized optimization framework.

**Fractional graph-based semi-supervised learning.** To improve the classification performance (in particular, fuzzy graphs and unbalanced labeled data) of G-SSL, fractional graph-based semi-supervised learning (De Nigris et al., 2017) embeds Lévy Flights into random walks on graphs by constructing from powers of the Laplacian matrix, i.e., the $L^{\gamma}$ operator. This operation can be used to generate different transition probabilities (i.e., corresponding to stochastic adjacency matrix) based on different $\gamma$ values. Intuitively, embedding Lévy Flights into random walks allows for better capturing mixing properties (i.e., dependence) in the data. Based on a fractional Laplacian matrix, $0 < \gamma < 1$, the anomalous (fractional) diffusion processes on networks can be constructed from the spectra data and eigenvectors of the Laplacian matrix. The fractional powers of $L$ allows Lévy random walks with long-range navigation on a network. For example, the long-range transitions

on a network can directly move node $u$ and node $v$ with the transition probability $m_{u \to v}^{(\gamma)}$ through a random walker, where $m_{u \to v}^{(\gamma)}$ is an element in the fractional transition matrix $\mathbf{M}^{(\gamma)}$ (see the graph within dotted circle in Figure 1). The transition probability $m_{u \to v}^{(\gamma)}$ between any two nodes whose geodesic distance is not infinite can be summarized as follow:

$$m_{u \to v}^{(\gamma)} = \delta_{uv} - \frac{(L^\gamma)_{uv}}{k_u^{(\gamma)}}, \tag{4}$$

where $\delta_{uv}$ represents the Kronecker delta, $k_u^{(\gamma)}$ denotes the fractional degree of the node $u$ and $k_u^{(\gamma)} \equiv (L^\gamma)_{uu}$. Eq. 4 provides transition probabilities for the Lévy Flights. Unlike the standard random walk, the Lévy Flights can jump immediately over several hops in a graph. This feature makes Lévy Flights a very good exploratory process. There is a price to pay for this: the typically sparse transition probability matrix becomes non-sparse. We can mitigate non-sparsity by taking a reasonable number of principal singular eigenvectors or limiting the number of terms in the Taylor expansion. Through replacing the $L$ operator with $L^\gamma = U^\top \Lambda^\gamma U$, the new optimization formulation leaves us with the following expression:

$$\min_F \left\{ 2F_{\cdot k}^T D_\gamma^{\sigma-1} L^\gamma D_\gamma^{\sigma-1} F_{\cdot k} + \mu (F_{\cdot k} - Y_{\cdot k})^T D_\gamma^{2\sigma-1} (F_{\cdot k} - Y_{\cdot k}) \right\}, \tag{5}$$

where $(D_\gamma)_{ii} = (L^\gamma)_{ii}$.

Let $0 < \gamma < 1$, the closed form solution for Eq. 5 can be obtain as fellow:

$$F_{\cdot k} = (1 - \alpha) \left( I - \alpha D_\gamma^{-\sigma} W_\gamma D_\gamma^{\sigma-1} \right)^{-1} Y_{\cdot k}, \tag{6}$$

for $k = 1, \ldots, K$. Therefore, we can conclude three particular fractional semi-supervised learning mehtods like G-SSL:

- Fractional SL method ($\sigma = 1$): $F_{\cdot k} = (1 - \alpha) \left( I - \alpha D_\gamma^{-1} W_\gamma \right)^{-1} Y_{\cdot k}$

- Fractional NL method ($\sigma = \frac{1}{2}$): $F_{\cdot k} = (1 - \alpha) \left( I - \alpha D_\gamma^{\frac{-1}{2}} W_\gamma D_\gamma^{\frac{-1}{2}} \right)^{-1} Y_{\cdot k}$

- Fractional PR method ($\sigma = 0$): $F_{\cdot k} = (1 - \alpha) \left( I - \alpha W_\gamma D_\gamma^{-1} \right)^{-1} Y_{\cdot k}$

### 3.2 PROPOSED FRACTIONAL GENERALIZED GRAPH CONVOLUTIONAL NETWORKS FOR SEMI-SUPERVISED NODE CLASSIFICATION

Although both G-SSL and fractional G-SSL achieve comparable and consistent (low variance) performance on some datasets, e.g., Les Miserables, Wikipedia-math, and MNIST, these approaches consider only the given adjacency matrix $W$ and the label matrix $Y$ without using the feature matrix $X$. Such limitation is crucial especially when dealing with datasets that not only exhibit a sophisticated topological graph structure but also provide node feature information, such as citation, biological, financial, and power grid networks. To address this drawback, through using the feature matrix $X$ instead of the label matrix $Y$ and encoding the graph structure by using neural network framework, many graph-based neural networks methods, e.g., graph convolutional networks (GCN), have been recently proposed and have been shown to demonstrate very impressive advances in semi-supervised learning performance on graphs. Next, we turn to discussing the new Fractional Generalized Graph Convolutional Networks (FGCN) for semi-supervised node classification.

**Fractional Generalized Graph Convolutional Networks (FGCN).** The key idea behind our proposed method is Fractional Generalized Sigma-based (FGS) filter $g_{FGS}(\alpha, \sigma, \gamma) = (1 - \alpha) \left( I - \alpha D_\gamma^{-\sigma} W_\gamma D_\gamma^{\sigma-1} \right)^{-1} = (1 - \alpha)(I - \alpha \tilde{L})^{-1}$. Let us insert Taylor series expansion into the expression of the FGS filter and this has the advantage of avoiding the inverse computation. We have the following expression:

$$g_{FGS}(\alpha, \sigma, \gamma) = (1 - \alpha) \left( I + (\alpha \tilde{L})^1 + (\alpha \tilde{L})^2 + (\alpha \tilde{L})^3 + \cdots \right) = (1 - \alpha) \sum_{i=0}^{\infty} \left( \alpha \tilde{L} \right)^i, \tag{7}$$

where $0 < \alpha \leq 1, 0 \leq \sigma \leq 1, 0 < \gamma \leq 1$. We then obtain the general classification function by multiplying the feature matrix $X$:

$$
\begin{aligned}
\bar{\mathcal{X}} &= g_{FGS}(\alpha, \sigma, \gamma)X \\
&= (1 - \alpha)\Big( \underbrace{X}_{\text{the first item}} + \underbrace{\alpha\tilde{L}X}_{\text{the second item}} + \underbrace{\alpha^2\tilde{L}^2 X}_{\text{the third item}} + \cdots \Big),
\end{aligned}
\tag{8}
$$

where, recursively,

$$
(X')_i = X + \alpha\tilde{L}(X')_{i-1}, \quad (X')_0 = X, \quad (i = 1, 2, \cdots).
\tag{9}
$$

**Convolutional layer.** During the training process, the convolutional model need to train the trainable parameters $(\mathbf{W}, \mathbf{b})$ of the graph filter which can scan the given input feature matrix into a series of feature maps with neurons. Thereby, we provide an implementation of Eq. 8 as a FGS convolutional layer:

$$
H^{(t+1)} = \sigma\left( (1 - \alpha) \sum_{i=0}^{\infty} \left(\alpha\tilde{L}\right)^i H^{(t)}\mathbf{W}^{(t)} \right),
\tag{10}
$$

where $H^{(t+1)}$ is the hidden layer output matrix of activations in the $t$-th layer and $H^{(0)} = X$, $\sigma(\cdot)$ is the adopted activation function, and $\mathbf{W}^t$ is the trainable weight in the $t$-th layer. Besides, we borrow the concept of the parallel system in reliability theory to improve the consistency of our proposed method. A parallel system is a configuration that the entire system works as long as not all involved components in the system fail. Hence, the parallel system structure is more robust against noisy inputs, compared to a single system structure.

**Lemma 1.** *Let $\bar{X}_{FGS}$ be the output matrix $P_1$ from a pooling layer. Let $\mathcal{U} = \{1, 2, \ldots, N\}$ be a finite population such that each unit $i, i \in \mathcal{U}$ is associated with an output matrix $X_{FGS}^{(i)}$, $i = 1, \ldots, N$. Then*

$$
\mathrm{Var}(\bar{X}_{FGS}) = \left(1 - \frac{n}{N}\right)\frac{S^2}{n}, \quad n < N, n \in \mathbb{Z}^+,
\tag{11}
$$

*where $S^2 = \sum_{i=1}^{N}\left(X_{FGS}^{(i)} - \bar{X}_{FGS}^U\right)^2/(N-1)$.*

*Suppose there are $n$ components in a parallel system, with the probability of non-failure $P_R^{(i)}$ (where $i = 1, \cdots, n$) in a parallel system, then the reliability of this parallel system $P_R^{PS}$ can be obtained with the following expression:*

$$
P_R^{PS} = 1 - \prod_{i=1}^{n}(1 - P_R^{(i)}) = 1 - (1 - P_R^{(1)}) \times (1 - P_R^{(2)}) \times \cdots \times (1 - P_R^{(n)}).
\tag{12}
$$

According to Lemma 11, the parallel system allows for enhancing stability and reducing estimation variance up to order of $n$ (i.e., $\mathrm{Var}(\bar{X}_{FGS}) = O(S^2/n)$). In this way, we establish both theoretical and practical guarantees for our proposed model to reach stable over a large set of hyperparameters, small datasets, and noisy labels based on this parallel implementation.

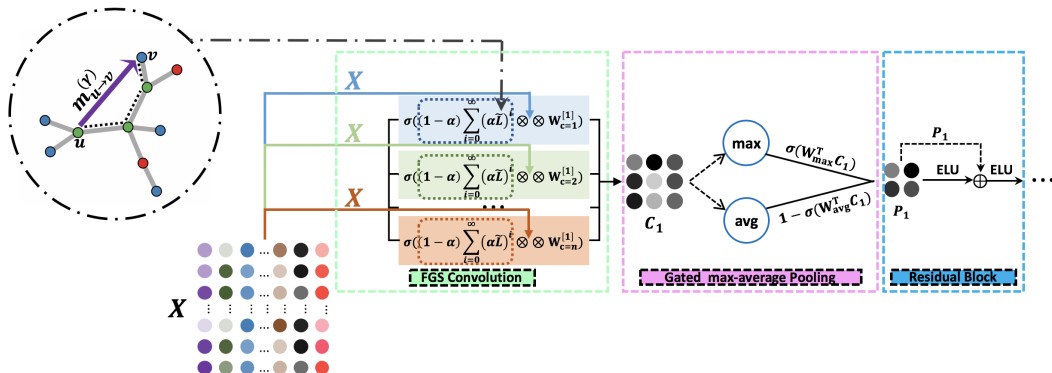

Figure 1: Illustration Fractional Generalized Graph Convolutional Network model. The input is the feature matrix $X$ and the graph within dotted circle represents embedding Lévy Flights into random walks on graph (where $L^\gamma$ is the Laplacian matrix $L$ to a power $\gamma$). FGCN architecture consists of three main components: (i) FGS convolutional layer with parallel structure; (ii) gated max-average pooling layer; (iii) activation block for residual learning.

**Pooling layer.** When implementing the form of pooling operation to aggregate information from the outputs of parallel FGS convolutional layer, instead of using some popular pooling functions such as max and average pooling, we apply the state-of-art pooling operation - *gated max-average pooling* (Lee et al., 2016) to capture the local and global information from all the nodes and graph structure. The rationale behind the *gated max-average pooling*, is that it considers "responsive" strategy (i.e., improving translation invariance and scale invariance via considering input in each gating mask) based on the *mixed max-average pooling* equation. We have:

$$f_{\text{gate}}(X_{FGS}) = \sigma\left(\mathbf{W}^\top X_{FGS}\right) f_{\max}(X_{FGS}) + \left(1 - \sigma\left(\mathbf{W}^\top X_{FGS}\right)\right) f_{\text{avg}}(X_{FGS}), \qquad (13)$$

where $\mathbf{W}$ is the trainable weight matrix, $X_{FGS}$ is the output matrix from the parallel FGS convolutional layer after concatenation operation.

**Residual building block.** Inspired by residual learning, we apply a residual block by adding the skip connection after the pooling layer. One of the advantages of the residual learning is the *identity* mapping which provides a direct path for propagating information. When using the residual building block, we adopt a similar scheme in He et al. (2016) to deal with the output of the pooling layer. Let $\mathcal{H}(x)$ be an underlying mapping and we cast it as $\mathcal{H}(x) = \mathcal{F}(x) + x$, where $\mathcal{F}(x)$ is the residual mapping defined by $\mathcal{H}(x) - x$. In other words, it is easier to optimize the residual mapping $\mathcal{F}(x)$ than optimizing the direct mapping $\mathcal{H}(x)$ and helps to avoid the gradient vanishing problem during training. To this end, we use an exponential linear unit (ELU) in direct mapping and place a rectified linear unit (ReLU) after addition in our model.

# 4 EXPERIMENTS

## 4.1 DATASETS

**Undirected networks.** Cora-ML (this Cora dataset consists of Machine Learning papers), Cite-Seer and PubMed are three standard citation networks benchmark datasets used for semi-supervised learning evaluation. In these citation networks, nodes represent publications, edges denote citation, the input feature matrix are bag of words and label matrix contain the class label of publication. We use the same data format as GCN, i.e., 20 labels per class in each citation network.

**Directed networks.** We evaluate our method on four directed networks - Cora, IEEE 118-bus system (IEEE bus), Texas 2000-bus system (TX bus), and South Carolina 500-bus system (SC bus). Unlike the small subset of the Cora-ML dataset (above), Cora is extracted from original dataset which contains more than $50,000$ computer science research papers, where nodes represent scientific papers and directed edges between node $u$ and node $v$ ($u \to v$) represent citation of paper $v$ in paper $u$. In addition to citation networks, we also test our method on power grid networks. For

IEEE 118-bus system, we consider a (unweighted-directed) graph as a model for the IEEE 118-bus system where nodes represent units such as generators, loads and buses, and edges represent the transmission lines. The input features of power grid network are generator active power upper bound (PMAX) and real power demand (PD) obtained from MATPOWER case struct. For Texas 2000-bus system and South Carolina 500-bus system, we treat them as weighted directed power grid networks. In particular, we use total line charging susceptance (BR_B) as edge weight and input features are: (i) real power demand (PD); (ii) reactive power demand (QD); (iii) voltage magnitude (VM); (iv) voltage angle (VA); (v) base voltage (BASE_KV). On directed networks, we test them separately from two scenarios: (i) on Cora, the number of features are decomposed into a low dimension (130 components) based on principal component analysis (PCA), the dataset is trained with 10% label rate and evaluated in another 10% fraction whose labels are withheld; (ii) on IEEE 118-bus system, Texas 2000-bus system, and South Carolina 500-bus system, 10% label rate for training set, 20% for validation and 70% for test sets. The statistics of data we used in the experimental section are summarized in Table 1.

Table 1: Dataset statistics.

| Dataset | Vertices | Edges | Features | Classes | Label rate | Train/Valid/Test |
|---|---|---|---|---|---|---|
| Cora-ML | 2,708 | 5,429 | 1,433 | 7 | 0.052 | 140 / 500 / 1,000 |
| CiteSeer | 3,327 | 4,732 | 3,703 | 6 | 0.036 | 120 / 500 / 1,000 |
| PubMed | 19,717 | 44,338 | 500 | 3 | 0.003 | 60 / 500 / 1,000 |
| Cora | 19,793 | 65,311 | 8,710 | 70 | 0.100 | 1,979 / 1,979 / 1,979 |
| IEEE Bus | 118 | 182 | 2 | 3 | 0.100 | 11 / 35 / 70 |
| TX Bus | 2,000 | 2,668 | 5 | 3 | 0.100 | 200 / 400 / 1,400 |
| SC Bus | 500 | 584 | 5 | 3 | 0.100 | 50 / 100 / 350 |

## 4.2 SETUP AND BASELINES

**Baselines.** In this setting, on undirected networks, we compare FGCN with the state-of-the art semi-supervised classification approaches including (i) using the label matrix as input: label propagation (LP) (Zhu & Ghahramani, 2002); (ii) using the feature matrix as input: DeepWalk (DW) (Perozzi et al., 2014), graph attention networks (GAT), GNN with ChebNet polynomials filter (ChebNet), GCN, GNNs with convolutional ARMA filters (ARMA), Graph Markov Neural Networks (GMNN) (Qu et al., 2019), and Large-Scale Learnable Graph Convolutional Networks (LGCNs) (Gao et al., 2018). In addition, on undirected networks, we use MotifNet (using symmetric motif adjacency matrix instead of Normalized Laplacian matrix in multivariate polynomial filters), ChebNet, GCN, ARMA, GMNN, and LGCNs as the benchmarks.

**Training setting details.** The training is done by using Adam optimizer with learning rate $lr_1 = 0.01$ for undirected networks and $lr_2 = \{0.1; 0.001\}$ for directed networks. To prevent the model from overfitting, we consider both adding dropout layer before two graph convolutional layers and kernel regularizers ($\ell_2$) in each layer. For undirected networks: we follow the experimental setup in Kipf & Welling (2017) to set the parameters of baselines (GAT, ChebNet, GCN, ARMA, GMNN, and LGCNs) by using two graph convolutional layers with 16 hidden units, $\ell_2$ regularization term with coefficient $5 \times 10^{-4}$, and dropout probability $p_{drop.}$ of 0.5 (for ARMA, except for number of hidden units, the hyperparameters setting are significantly different from others). For directed networks: we consider the MotifNet setting with using dropout rate $p_{drop.}$ of 0.5, $\ell_2$ regularization term with coefficient 0.001, and the degree of multivariate polynomials $k = 4$ for MotifNet; for other three baselines (i.e., ChebNet, GCN, ARMA, GMNN, and LGCNs), we also use a two-layer network but with 16 hidden units, learning rate $lr_2$ with 0.001, 0.5 dropout rate and $\ell_2$ regularization weight of $5 \times 10^{-4}$. In the following, Table 2 displays the best hyperparameter configurations of FGCN for each dataset by using standard grid search mechanism (the optimal kernel regularization weight $\ell_2$ always equal to $5 \times 10^{-4}$).

Table 2: Hyperparameters setting of FGCN

| Dataset | $p_{drop.}$ | $n_h^{[1]}$ | $\alpha$ | $\sigma$ | $\gamma$ | $n_{FGS}$ | $\{\#\mathcal{C}^{[1]}; \#\mathcal{C}^{[2]}\}$ | $\phi(RB)$ |
|---------|-------------|-------------|----------|----------|----------|-----------|---------------------------------------------|------------|
| Cora-ML | 0.75 | 32 | 0.6 | 0.50 | 1.000 | 7 | {5;2} | ELU + ReLU |
| CiteSeer | 0.75 | 32 | 0.6 | 0.53 | 1.000 | 7 | {3;2} | ELU + ELU |
| PubMed | 0.00 | 32 | 0.6 | 0.50 | 1.000 | 7 | {5;2} | ELU + ReLU |
| Cora | 0.00 | 128 | 0.4 | 1.00 | 1.000 | 15 | {1;1} | ELU + ELU |
| IEEE Bus | 0.00 | 128 | 0.1 | 0.20 | 0.004 | 40 | {7;2} | ELU + ELU |
| TX Bus | 0.00 | 128 | 0.3 | 0.10 | 0.100 | 14 | {5;2} | ELU + ELU |
| SC Bus | 0.00 | 128 | 0.2 | 0.10 | 0.100 | 20 | {5;2} | ELU + ELU |

* $n_h^{[1]}$ denotes the number of hidden units in the first layer; $\#\mathcal{C}^{[1]}$ and $\#\mathcal{C}^{[2]}$ represents the number of statistically independent parallel components in the first layer and the second layer respectively; $n_{FGS}$ denotes the number of items in Taylor series polynomial expansion of FGS filter; $\phi(RB)$ represents the type of activation functions in residual block structure.

## 5 RESULTS

**Performance comparison of semi-supervised node classification.** Table 3 reports the average accuracy delivered by FGCN and competing methods for fixed train/valid/test sizes (see Table 1). The best performance for each dataset is marked in bold.

Table 3 shows that FGCN outperforms all competing approaches in all data sets, except for CiteSeer (FGCN delivers the fouth best accuracy result), PubMed (FGCN delivers the second best accuracy result), Cora (FGCN delivers the third best accuracy result after the LGCNs approach and the difference between FGCN and LGCNs for Cora is less than 0.05%). The improvement gain of FGCN over the next most accurate method ranges from 0.75% (for undirected Cora-ML over GMNN) to 4.21% (for directed IEEE 118-Bus over GMNN). Remarkably, methods that are applicable both to undirected and directed networks (i.e., ChebNet, GCN, ARMA, GMNN, and LGCNs) tend to deliver noticeably lower accuracy results for a directed networks (especially on weighted-directed networks), while the new FGCN method yields a more stable performance across both directed and undirected networks. In turn, CiteSeer (unweighted-undirected), GMNN and LGCNs outperform FGCN up to 1.7%. For the PubMed (unweighted-undirected), our FGCN model delivers marginally better performance results than LGCNs, and GMNN has better performance than FGCN. Finally, both GMNN and LGCNs outperform FGCN on CORA (unweighted-directed), but the delivered accuracy results are very close. Based on the obtain results, the new FGCN approach tends to be the most competitive and, hence, preferred node classification method for sparser networks with higher label rates.

Furthermore, the IEEE 118-Bus data set is the smallest among the considered data (see Table 1), and we might expect to observe lower accuracy results for this data set due to a limited training set. However, the accuracy yielded by FGCN is among the highest ones across all data sets.

Table 3: Comparison of average accuracy (%) of semi-supervised classification approaches for both undirected networks and directed networks.

| Method | Undirect networks | | | Directed networks | | | |
|--------|---------|----------|--------|------|----------|--------|--------|
| | **Cora-ML** | **CiteSeer** | **PubMed** | **Cora** | **IEEE Bus** | **TX Bus** | **SC Bus** |
| LP | 68.70 | 46.32 | 65.92 | - | - | - | - |
| DW | 67.20 | 43.27 | 65.33 | - | - | - | - |
| MotifNet | - | - | - | 60.00 | 65.75 | 82.00 | 95.18 |
| GAT | 83.11 | 70.85 | 78.56 | - | - | - | - |
| ChebNet | 81.45 | 70.23 | 78.40 | 58.93 | 60.00 | 80.04 | 94.13 |
| GCN | 81.50 | 71.11 | 79.00 | 57.75 | 52.86 | 73.36 | 90.33 |
| ARMA | 82.80 | 72.30 | 78.80 | 58.99 | 70.55 | 81.20 | 94.33 |
| GMNN | 83.72 | **73.10** | **81.80** | **61.20** | 78.88 | 86.21 | 96.57 |
| LGCNs | 83.35 | 73.08 | 79.51 | 60.72 | 71.43 | 85.57 | 95.14 |
| FGCN (ours) | **84.35** | 71.89 | 79.56 | 60.70 | **82.20** | **87.74** | **97.61** |

**Evaluation of FGCN-specific parameters.** During grid search over three parameters (i.e., $\alpha$, $\sigma$, and $\gamma$), we find that: (i) the regularization parameter $\alpha$ which used to specify the relative importance of a graph in clustering strongly relates to the probability of initial conditions for random walks when the self-refreshing process works, and it strongly influences the network's generalization ability and node classification performance for all datasets; (ii) the free unifying parameter $\sigma$ provides enough flexibility to construct a canonical formulation of different graph-based semi-supervised methods. Table 3 indicates that the optimal value of $\sigma$ depends on both the types of networks (undirected and directed) and label rate not on the size of network; (iii) the fractional power parameter $\gamma$ substantially impacts the accuracy of node classification for the small datasets (see e.g., Figure 2), however, no similarly strong influence is found in the larger datasets.

**Sensitivity analysis.** In the sensitivity analysis setting, we have the ability to analyze the sensitivity of the node classification accuracy to variation from three FGCN-specific parameters - $\alpha \in \{0.1, \cdots, 1\}$, $\sigma \in \{0, 0.1, \cdots, 1\}$, and $\gamma \in \{0.001, 0.01, 0.1, 1\}$. In this case, we only show the results from sensitively analysis for FGCN model on IEEE 118-bus dataset. First, we perform the parameter learning experiments on four scenarios with a fixed parameter $\gamma$. Figure 2 shows that the accuracy substantially decreases when $\alpha$ is larger than 0.8, especially in $\gamma$ equals to 0.001 and 0.01 (see Figure 2a, 2b). Setting $\gamma = \{0.1, 1\}$, we observe that the classification accuracy nearly monotonic decreases while increasing $\alpha$. Additionally, there is little difference of classification accuracy between $\gamma = 0.001$ and $\gamma = 0.01$ and FGCN generally gives consistent and higher accuracy when the $\alpha$ parameter is within the range of $\{0.1, 0.2, 0.3, 0.4\}$. We then explore the variation of accuracy based on tuning parameter $\gamma$ within the range of $[0.001, 0.002, \cdots, 0.01]$ (setting $\sigma \in \{0, 0.1, \cdots, 1\}$ at the same time), however, it is hard to obtain the optimal $(\hat{\sigma}, \hat{\gamma})$ combination through gathering finite experimental results (100 runs) since some of the results are very close. Therefore, we run the following experiments to demonstrate the impact evaluation of $\gamma$:

- *Step 1:* Setting $\gamma \in \{0.001, 0.002, \cdots, 0.1\}$.

- *Step 2:* For each $\gamma$ (i.e., fix the $\gamma$ value in each experiment), we run our proposed model 100 times separately for $\sigma$ from 0 to 1 by 0.01; Then we can obtain the 100 average accuracies for each $\gamma$ under fixed $\gamma$, that is, $\{AAC_{\sigma=0}, ACC_{\sigma=0.01}, \cdots, AAC_{\sigma=1}\}^{[i]}$, where $i = 1, \cdots, 10$.

- *Step 3:* Fitting the Gaussian distribution to $\{AAC_{\sigma=0}, ACC_{\sigma=0.01}, \cdots, AAC_{\sigma=1}\}^{[i]}$ (see Figure 3b).

Similar to $\gamma$, we can fit the Gaussian distribution to $\{AAC_{\gamma=0.001}, ACC_{\gamma=0.002}, \cdots, AAC_{\gamma=0.01}\}^{[j]}$ by fixing $\sigma$, where $j = 1, \cdots, 11$ (see Figure 3a).

From Figure 3, we find that there exist larger difference between the shapes of approximate Gaussian distributions by fixing the parameter $\sigma$ than fixing the parameter $\gamma$ which means the parameter $\sigma$ is more important factor in FGCN approach for small datasets.

## 6 CONCLUSION

In this paper we have proposed a new Fractional Generalized Graph Convolutional Networks (FGCN) method to semi-supervised learning on graphs that enables to better capture the intrinsic local graph topology. The key idea behind our new approach is a new fractional generalized graph-based convolutional filter which casts the Lévy Flights into random walks on graphs and, as a result, allows for a more efficient, accurate and robust exploration of the local graph structure which often plays the key role in graph learning performance, especially for heterogeneous graphs.

Our numerical studies have indicated that the new FGCN method tends to outperform all competing deep learning approaches on both unweighted-directed and unweighted-undirected graphs in all considered data sets, except of Cora for which the difference between the best result delivered by the GMNN and FGCN is less than 0.85%. The gain in learning accuracy of FGCN over the next best competitor ranges from 0.75% to 4.21%, and the highest gain has been achieved for the IEEE 118-Bus data set which is the smallest among the considered data sets. Furthermore, in contrast to the competing approaches, FGCN tends to deliver a more stable performance across directed and undirected networks regardless of the label rate.

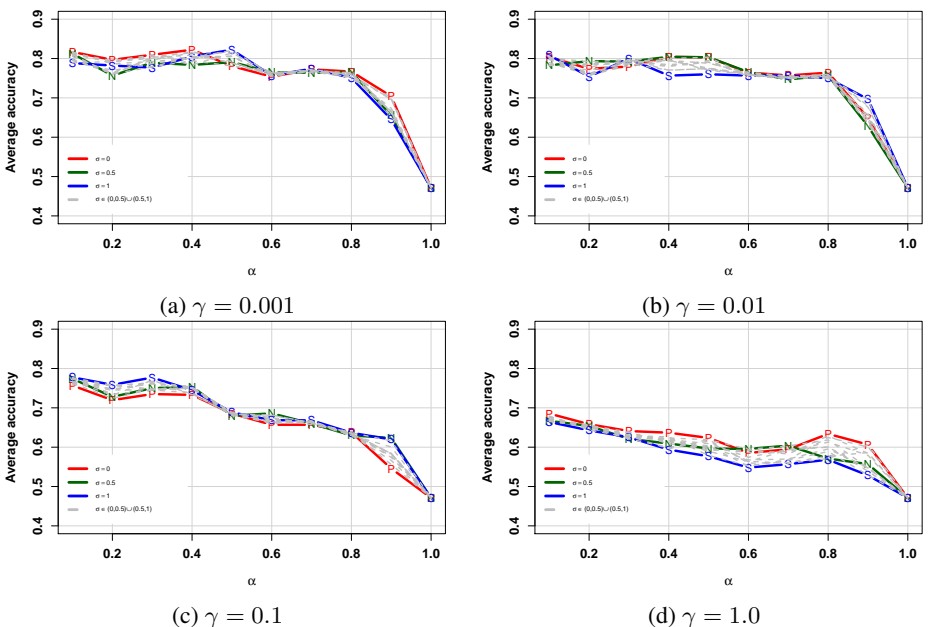

Figure 2: Accuracy of FGCN for PageRank, Normalized Laplacian, and Standard Laplacian, which depends on the function of $\alpha$ and fixes the fractional power $\gamma = \{0.001$ (a), $0.01$ (b), $0.1$ (c), $1.0$ (d)$\}$

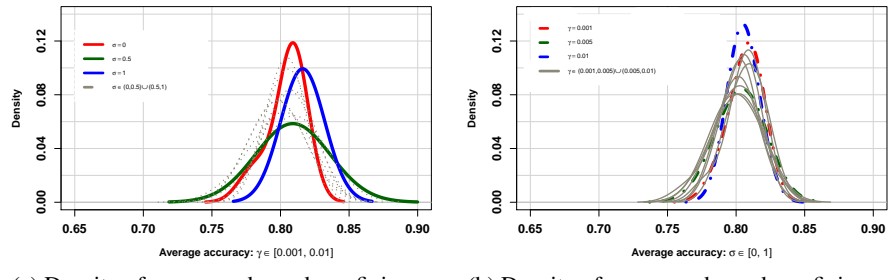

(a) Density of accuracy depends on fixing $\sigma$    (b) Density of accuracy depends on fixing $\gamma$

Figure 3: Generalized Gaussian density of accuracy of FGCN: (a) the solid curves represent three particular scenarios in FGCN, where the red curve denotes the PR-based method ($\sigma = 0$), green curve denotes the NL-based method ($\sigma = 0.5$), and blue curve denotes the SL-based method ($\sigma = 1$); the dotted curves represent common scenarios in FGCN model, that is, the generalized parameter $\sigma$ falls within the range of $(0, 0.5) \cup (0.5, 1)$. (b) the red, green, blue dotted curve represents the scenario when setting the fractional parameter $\gamma$ equals to 0.001, 0.005, and 0.01, respectively; The solid curves represent scenarios where $\gamma$ in the range of $(0.001, 0.005) \cup (0.005, 0.01)$.

In the future we plan to advance the proposed FGCN technique to learning on multilayer networks and to enhance graph learning process with topological information on the underlying deep neural network.

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

## A APPENDIX

**Proof of Lemma 1:** The parallel structure (Henley & Kumamoto, 1981) is constructed in a similar spirit as bagging of features in random forest and other ensemble learning methods. The key rationale behind the parallel structure is to reduce variance and increase stability.

Let us first consider (11). Suppose that $\Upsilon$ is a random sample drawn without replacement from a finite population $\mathcal{U} = \{1, 2, \ldots, N\}$ according to a sampling design $p(\cdot)$. Each unit $i, i = 1, \ldots, N$ of $\mathcal{U}$ is associated with $X_{FGS}^{(i)}$, i.e. the $i$-th output matrix from FGS convolution for new feature matrix $X_i$. Probability of choosing sample $\upsilon$ is $Pr(\Upsilon = \upsilon) = p(\Upsilon) > 0$ for all $\upsilon \in U$ and $\upsilon \neq \emptyset$. Let $Z$ be the indicator variable such that

$$Z_i = \begin{cases} 1, & \text{if } X_i \text{ is in the sample} \\ 0, & \text{otherwise.} \end{cases} \tag{14}$$

Hence, probability that the unit $i, i = 1, \ldots, N$ is selected, is given by $\pi_i = E(Z_i)$, and probability that units $i$ and $j$, $i, j = 1, \ldots, N$ are selected simultaneously is $\pi_{ij} = E(Z_i Z_j)$; $\pi_i$ and $\pi_{ij}$ are called *first order inclusion probability* and *second order inclusion probability*, respectively.

Since in the current paper, we consider a simple random sampling design of $n$ units $X_{FGS}^{(i)}$ without replacement from $\mathcal{U}$, $\pi_i = n/N$ and $\pi_{ij} = n(n-1)/N(N-1)$.

Let $\bar{X}_{FGS} = \sum_{i \in N} X_{FGS}^{(i)}/n = \sum_{i=1}^{N} Z_i X_{FGS}^{(i)}/n$. Hence, given $\{X_{FGS}^{(1)}, \ldots, X_{FGS}^{(N)}\}$, we find that

$$E[\bar{X}_{FGS}] = \sum_{i=1}^{N} E[Z_i] \frac{X_{FGS}^{(i)}}{n} = \sum_{i=1}^{N} \pi_i \frac{X_{FGS}^{(i)}}{n} = \sum_{i=1}^{N} \frac{n}{N} \frac{X_{FGS}^{(i)}}{n} = \sum_{i=1}^{N} \frac{X_{FGS}^{(i)}}{N} = \bar{X}_{FGS}^{U},$$

where $\bar{X}_{FGS}$ is be the output matrix $P_1$ from a pooling layer (see Figure 1), which implies that $\bar{X}_{FGS}$ is an unbiased estimator of population mean $\bar{X}_{FGS}^U$.

In turn,

$$
\begin{aligned}
\operatorname{Var}(\bar{X}_{FGS}) &= \frac{1}{n^2} V \left( \sum_{i=1}^{N} Z_i X_{FGS}^{(i)} \right) \\
&= \frac{1}{n^2} \operatorname{Cov} \left( \sum_{i=1}^{N} Z_i X_{FGS}^{(i)}, \sum_{j=1}^{N} Z_j X_{FGS}^{(j)} \right) \\
&= \frac{1}{n^2} \left[ \sum_{i=1}^{N} (X_{FGS}^{(i)})^2 \operatorname{Var}(Z_i) + \sum_{i=1}^{N} \sum_{j \neq i}^{N} X_{FGS}^{(i)} X_{FGS}^{(j)} \operatorname{Cov}(Z_i, Z_j) \right] \\
&= \frac{1}{n^2} \left[ \frac{n}{N} \left( 1 - \frac{n}{N} \right) \sum_{i=1}^{N} (X_{FGS}^{(i)})^2 - \sum_{i=1}^{N} \sum_{j \neq i}^{N} X_{FGS}^{(i)} X_{FGS}^{(j)} \frac{1}{N-1} \left( 1 - \frac{n}{N} \right) \left( \frac{n}{N} \right) \right] \\
&= \frac{1}{n} \left( 1 - \frac{n}{N} \right) \frac{1}{N(N-1)} \left[ (N-1) \sum_{i=1}^{N} (X_{FGS}^{(i)})^2 - \left( \sum_{i=1}^{N} X_{FGS}^{(i)} \right)^2 + \sum_{i=1}^{N} (X_{FGS}^{(i)})^2 \right] \\
&= \frac{1}{n} \left( 1 - \frac{n}{N} \right) \frac{1}{N(N-1)} \left[ N \sum_{i=1}^{N} (X_{FGS}^{(i)})^2 - \left( \sum_{i=1}^{N} X_{FGS}^{(i)} \right)^2 \right] \\
&= \left( 1 - \frac{n}{N} \right) \frac{S^2}{n},
\end{aligned}
\tag{15}
$$

where $S^2 = \sum_{i=1}^{N} \left( X_{FGS}^{(i)} - \bar{X}_{FGS}^U \right)^2 / (N-1)$ and the factor $(1 - n/N)$ is called the finite population correction (FPC). Hence, the proposed PS structure decreases an original estimation variance $S$ by order of $n$. Note that we can consider other sampling designs $p(\cdot)$, including, for example, weighted sampling and $p$-extended simple random sampling with replacement (see discussion by Wong & Easton, 1980; Scott & Köhl, 1994; Ozturk & Balakrishnan, 2019, and references therein).

The stability result of the parallel structure (Nowak & Collins, 2012) follows from verbatim application of the probability bound on the intersection of independent events.

## B APPENDIX

The general idea of graph-based semi-supervised learning (G-SSL) is based on two widely used optimization frameworks. The first formulation, the Standard Laplacian based formulation (Zhou & Burges, 2007) as follows:

$$
\min_F \left\{ \sum_{i=1}^{N} \sum_{j=1}^{N} w_{ij} \| F_{i.} - F_{j.} \|^2 + \mu \sum_{i=1}^{N} d_i \| F_{i.} - Y_{i.} \|^2 \right\},
\tag{16}
$$

where $d_{ii}$ is $(i, i)$-element in degree matrix $D$ and $w_{ij}$ represents the edge weight for edge $e_{ij}$ in adjacency matrix $W$. For the second formulation, the Normalized Laplacian based formulation (Zhou et al., 2004), is as follows:

$$
\min_F \left\{ \sum_{i=1}^{N} \sum_{j=1}^{N} w_{ij} \left\| \frac{F_{i.}}{\sqrt{d_{ii}}} - \frac{F_{j.}}{\sqrt{d_{jj}}} \right\|^2 + \mu \sum_{i=1}^{N} \| F_{i.} - Y_{i.} \|^2 \right\}
\tag{17}
$$

The following lemma (Avrachenkov et al., 2012) asserts that the generalized optimization framework, i.e., G-SSL, which has as particular cases the two above mentioned formulations:

**Lemma 2.** *Let $\sigma$ denote an alternative parameter on the power of degree matrix $D$ whose entries are the degrees $d_{ii}$; and let $0 \leq \sigma \leq 1$. Then*

$$\min_{F} \left\{ \sum_{i=1}^{N} \sum_{j=1}^{N} w_{ij} \left\| d_{ii}^{\sigma-1} F_{i.} - d_{jj}^{\sigma-1} F_{j.} \right\|^2 + \mu \sum_{i=1}^{N} d_{ii}^{2\sigma-1} \left\| F_{i.} - Y_{i.} \right\|^2 \right\}.$$

The classification functions for the generalized semi-supervised learning are given by

$$F_{.k} = (1 - \alpha) \left( I - \alpha D^{-\sigma} W D^{\sigma-1} \right)^{-1} Y_{.k}.$$

**Proof:** The objective function of the generalized semi-supervised learning framework (i.e., Lemma 2.) can be rewritten in the following matrix form:

$$Q(F) = 2 \sum_{k=1}^{K} F_{.k}^T D^{\sigma-1} L D^{\sigma-1} F_{.k} + \mu \sum_{k=1}^{K} (F_{.k} - Y_{.k})^T D^{2\sigma-1} (F_{.k} - Y_{.k}).$$

Given the first order optimality condition $D_{F.k} Q(F) = 0$, we have

$$2 F_{.k}^T \left( D^{\sigma-1} L D^{\sigma-1} + D^{\sigma-1} L^T D^{\sigma-1} \right) + 2 \mu (F_{.k} - Y_{.k})^T D^{2\sigma-1} = 0.$$

Multiplying the above expression from the right hand side by $D^{-2\sigma+1}$ leads to

$$2 F_{.k}^T \left( D^{\sigma-1} \left( L + L^T \right) D^{-\sigma} \right) + 2 \mu (F_{.k} - Y_{.k})^T = 0.$$

Then, substituting $L = D - W$ and rearranging the terms yields

$$F_{.k}^T \left( 2I - D^{\sigma-1} \left( W + W^T \right) D^{-\sigma} + \mu I \right) - \mu Y_{.k}^T = 0.$$

Since the resulting adjacency matrix $W$ is a symmetric matrix (see transformation 3 in the main body of the paper through replacing $\mu$ with $\alpha = 2/(2 + \mu)$), we obtain

$$F_{.k}^T = \mu Y_{.k}^T \left( 2I - 2 D^{\sigma-1} W D^{-\sigma} + \mu I \right)^{-1},$$

which concludes the proof.

