# OpenReview forum: "Fractional Graph Convolutional Networks (FGCN) for Semi-Supervised Learning"
_ICLR.cc/2020/Conference — Reject_

### Official Review · AnonReviewer1 · 2019-10-23
**Official Blind Review #1**

**Rating:** 6

**Review:**

This paper presents a fractional generalized graph convolutional networks for semi-supervised learning. The authors design a new graph convolutional filter based on Levy Flights, and propose new feature propagation rules on graphs. Experimental results on multiple graph datasets are reported and discussed.

Pros.
1. This paper presents a nice overview of three popular semi-supervised learning methods in Section 3.1, and presents insights regarding these models.
2. A new graph convolutional filter is proposed. The motivation is clear, and the technical details are easy to follow.
3. Experiments on five benchmark datasets are conducted. Both undirected and directed graphs are used in experiments.

Cons.
1. The proposed method contains three major components: parallel FGS convolution, pooling, and residual block. Although some justifications are provided for such designs, it is difficult to justify the role of each component. In other words, it is unclear whether the performance gain is from the parallel structure, or the residual block. What would be the model performance without parallel structures? Also, given that FGCN has quite a few layers, the motivation of using residual blocks should be carefully justified.
2. Many recent methods on graph neural networks are not discussed or included as baselines, such as [a-b].
[a] GMNN: Graph Markov Neural Networks, ICML 2019
[b] Large-Scale Learnable Graph Convolutional Networks, KDD 2018
[c] SPAGAN: Shortest Path Graph Attention Network, IJCAI 2019


-------------------------------------------------
The response from authors addressed many of my concerns. The rating has been updated.

**Experience Assessment:**

I have published one or two papers in this area.

**Review Assessment: Checking Correctness Of Derivations And Theory:**

I assessed the sensibility of the derivations and theory.

**Review Assessment: Checking Correctness Of Experiments:**

I carefully checked the experiments.

**Review Assessment: Thoroughness In Paper Reading:**

I read the paper thoroughly.

---

### Official Review · AnonReviewer3 · 2019-10-23
**Official Blind Review #3**

**Rating:** 3

**Review:**

This paper proposes a fractional graph convolutional networks for semi-supervised learning. The proposed method used a classification function of a fractional graph semi-supervised learning (GSSL) [De Nigris et al., 2017] as a graph filter. In addition, the authors adopt a parallel system and weighted combination of max and average pool. Experimental results show that the proposed method (FGCN) shows the best accuracy compared to other recent graph-based neural networks for all datasets except one.

The key approach of the proposed method is to apply a classification function (equation (3)) obtained by solving a GSSL problem to graph convolutional networks. However, this idea is too incremental and applying the classification function to graph filter is very trivial. This works also combines the fractional GSSL with a parallel system and weighted pool. But, it is not clear which contribution actually improves the results. Moreover, the intuition of the fractional approach is not clear too, e.g., how the optimization (equation (2)) is derived?, and some explanations are unnatural to demonstrate the methodology, e.g., equation (4). For these reasons, this paper is under the bar of acceptance.

Main concerns:
1. What is the intuition of the optimization of GSSL? How is it obtained? And among all fractional methods (e.g., SL, NL, and PR), which one doe achieve the best performance?

2. The FGS filter in equation (7) is the sum of infinite terms. However, in practical, it is impossible to compute the infinite terms. Does this approximate the sum of finite terms? If does, what is the number of truncation?

3. The authors mention that they establish a theoretical guarantee of the parallel system. But, I could not find any theoretical results. It would be better to include the analysis in the paper.


Minor concerns:
1. In page 3, please edit “forulation” -> “formulation”
2. In equation (8), I think “X+\alpha \tilde{L} X” should change to “\tilde{L} X”


**Experience Assessment:**

I have read many papers in this area.

**Review Assessment: Checking Correctness Of Derivations And Theory:**

I assessed the sensibility of the derivations and theory.

**Review Assessment: Checking Correctness Of Experiments:**

I assessed the sensibility of the experiments.

**Review Assessment: Thoroughness In Paper Reading:**

I read the paper at least twice and used my best judgement in assessing the paper.

---

### Official Review · AnonReviewer2 · 2019-10-29
**Official Blind Review #2**

**Rating:** 8

**Review:**

The paper provides a new model for semi-supervised node classification in directed and undirected graphs. It is based on a novel fractional filter for graph conv networks, which generalizes several previously employed graph semi-supervised learning frameworks, by introducing a fractional hyperparameter (sigma in the paper), using fractional powers of the Laplace operator.

The relevant previous work in the area seems to be cited, and the paper appropriately embedded in the previous work

Empirically, the method outperforms several established baseline models (classical and neural) on standard datasets in the node classification task. in particular one with a low number of labeled nodes. A sensitivity analysis is performed to assess the impact of the hyperparameters of the FGCN.

I believe the experimental results could justify an accept, but I would not claim I am an expert in semi-supervised learning on graphs.


Questions:

How can the architecture be extended to handle edge types?

How were the hyperparameters of the baselines tuned?

Small things:
Could you provide more context around the reliability in parallel systems (eq 11)? It is not clear how this relates to the rest of the paper.

Please add a citation for gated max average pooling.


**Experience Assessment:**

I have read many papers in this area.

**Review Assessment: Checking Correctness Of Derivations And Theory:**

I did not assess the derivations or theory.

**Review Assessment: Checking Correctness Of Experiments:**

I assessed the sensibility of the experiments.

**Review Assessment: Thoroughness In Paper Reading:**

I read the paper at least twice and used my best judgement in assessing the paper.

---

### Author Response · Authors · 2019-09-30
**Some Corrections**

There is a typo (method name) in our paper, we correct it - it should be L\'evy Flights rather than L\'evy Fights. Besides, we also correct some other typos in "Training setting details" part. Here is a dropbox link for the corrected version and code: https://www.dropbox.com/sh/ajtz6inf677nkcv/AACXkFRZjRrCkxYkxJDNfks0a?dl=0.

Thanks!

---

### Decision · Program_Chairs · 2019-12-19

**Decision:**

Reject

**Comment:**

This paper proposes a fractional graph convolutional networks for semi-supervised learning, using a classification function repurposed from previous work, as well as parallelization and weighted combinations of pooling function. This leads to good results on several tasks.
Reviewers had concerns about the part played by each piece, the lack of comparison to recent related work, and asked for better explanation of the rationale of the method and more experimental details. Authors provided explanations and details, and a more thorough set of comparison to other work, showing better performance in some but not all cases.
However, concerns that the proposed innovations are too incremental remain.
Therefore, we cannot recommend acceptance.